# Manipulation of Meiotic Recombination to Hasten Crop Improvement

**DOI:** 10.3390/biology11030369

**Published:** 2022-02-25

**Authors:** Ian Fayos, Julien Frouin, Donaldo Meynard, Aurore Vernet, Léo Herbert, Emmanuel Guiderdoni

**Affiliations:** 1Meiogenix, 38 rue Sevran, 75011 Paris, France; ian.fayos@cirad.fr (I.F.); leo.herbert@cirad.fr (L.H.); 2CIRAD, UMR AGAP Institut, F-34398 Montpellier, France; julien.frouin@cirad.fr (J.F.); donaldo.meynard@cirad.fr (D.M.); aurore.vernet@cirad.fr (A.V.); 3UMR AGAP Institut, Université de Montpellier, CIRAD, INRAE, Institut Agro, F-34398 Montpellier, France

**Keywords:** breeding, crops, engineering, meiosis, recombination

## Abstract

**Simple Summary:**

Harnessing the natural and induced diversity existing in plant genetic resources is fundamental for building future crops more sober in fertilizers, water, and pesticides that can cope with climate instability while yielding healthier and more nutritious products. The essence of plant breeding is to combine favorable traits in crossing parental varieties to select novel performing associations amongst the progenies. These associations are the product of recombination between the parental chromosomes occurring during meiosis, mainly by a reciprocal DNA exchange called Cross Over (CO). However, recombination does not occur randomly along the chromosomes, and COs are limited in number often hampering the desired associations of favorable traits. This review surveys the recent advances in methods for achieving a stimulation and/or a redistribution of meiotic COs along the parental chromosomes and targeting COs specifically at desired chromosomal sites.

**Abstract:**

Reciprocal (cross-overs = COs) and non-reciprocal (gene conversion) DNA exchanges between the parental chromosomes (the homologs) during meiotic recombination are, together with mutation, the drivers for the evolution and adaptation of species. In plant breeding, recombination combines alleles from genetically diverse accessions to generate new haplotypes on which selection can act. In recent years, a spectacular progress has been accomplished in the understanding of the mechanisms underlying meiotic recombination in both model and crop plants as well as in the modulation of meiotic recombination using different strategies. The latter includes the stimulation and redistribution of COs by either modifying environmental conditions (e.g., T°), harnessing particular genomic situations (e.g., triploidy in Brassicaceae), or inactivating/over-expressing meiotic genes, notably some involved in the DNA double-strand break (DSB) repair pathways. These tools could be particularly useful for shuffling diversity in pre-breeding generations. Furthermore, thanks to the site-specific properties of genome editing technologies the targeting of meiotic recombination at specific chromosomal regions nowadays appears an attainable goal. Directing COs at desired chromosomal positions would allow breaking linkage situations existing between favorable and unfavorable alleles, the so-called linkage drag, and accelerate genetic gain. This review surveys the recent achievements in the manipulation of meiotic recombination in plants that could be integrated into breeding schemes to meet the challenges of deploying crops that are more resilient to climate instability, resistant to pathogens and pests, and sparing in their input requirements.

## 1. Introduction

In order to tackle the major challenges faced by agriculture in the next decades, plant breeding shall contribute by building novel crops resilient to climate change, resistant to pathogen and pest outbreaks, more resource-use efficient, and optimizing beneficial biological interactions. In that aim, plant breeding needs to tap and shuffle alleles existing in both natural and induced diversity to create new, agronomically favorable allele combinations.

Allele shuffling between the parental chromosomes, the homologs, is the major outcome of meiotic recombination. However, shuffling can be hampered when favorable genes/alleles reside in chromosomal regions that are not or poorly amenable to recombination. This holds particularly true for chromosomal regions exhibiting high interhomolog sequence divergence or structural variation in crosses. These divergent regions may contain important genes like some contributing to the dispensable genome, i.e., existing in only a subset of accessions, that represents a crucial gene reservoir for adaptation of crops to global change. Importance of the dispensable genome has recently been illustrated in rice by the isolation of genes underlying crucial adaptive mechanisms (e.g., submergence survival and avoidance, phosphorus uptake, fungal pathogen resistance) [1,2]. Recombination limitation due to sequence divergence/structural variation is also a bottleneck to combine the genomes of crops with those of their wild relatives and harness their potential in breeding. Furthermore, favorable genes/alleles can be tightly genetically linked to genes with alleles having an unfavorable action, a phenomenon called linkage drag that also hampers transfer of the former genes to elite materials. Therefore, gaining a better understanding of the mechanisms underlying meiotic recombination and developing tools for its modulation and targeting are of primary importance for improving plant breeding efficiency and accuracy.

## 2. Control of Meiotic Recombination

Aside from being limited in number, cross-overs (COs) exhibit neither an even nor a random distribution along the chromosomes. Long Terminal Repeat (LTR) transposable elements (TE)-rich heterochromatic regions, that can account for a large fraction of the genome in some crop species such as wheat, barley or maize, and centromeric regions are generally devoid of COs, despite a widespread occurrence of meiotic DSBs as shown by the sequencing of meiotic RAD51-bound ssDNA [3]. Euchromatic regions, rich in actively transcribed genes and DNA transposons, are generally prone to COs. In Arabidopsis, 80% of the COs occur in 26% of the genome [4] and the highest CO frequencies deduced from F2 population analyses, are observed in pericentromeric regions [5]. In rice, as in many plant species, an increasing gradient is observed from the proximal (centromeric) to the distal (telomeric) regions of the chromosomes, with exception of some chromosome arms [6] (Figure 1). The fact that COs exhibit an uneven distribution along the chromosomes in plants, limits access to genes of agronomical relevance located in “cold” recombination regions, that can represent a large fraction of the genome of crop species. For instance, in bread wheat chromosome 3B, 90% of the COs are located in 14% of the chromosome physical length while important genes are embedded in heterochromatic, LTR-TE-rich regions [7].

Meiotic recombination is initiated during the prophase of the first division of meiosis, by the self-infliction of several hundreds of chromosomal DNA double-strand breaks (DSBs) by the SPO11 catalytic complex in microspore- and megaspore- mother cells. Induction of DSBs triggers the pairing of homologous chromosomes, the parental homologs.

DSBs have to be repaired by different mechanisms. Only a sub-fraction of the DSBs—typically 1–3 per chromosome pair in most Eukaryotes [8]—is repaired to produce a reciprocal exchange of DNA, the CO, between the homologs (Figure 2). Endo-nucleolytic cleavage liberates SPO11 that remains covalently attached to a short oligonucleotide at each 5′ strand of both DSB ends [9]. The 5′ strand is further 5′-3′ resected by an exonuclease to generate long single-stranded DNA tails on either side of the DSB, coated by the RPA complex. Following RPA replacement by the RAD51 and DMC1 recombinases, this nucleoprotein filament searches and invades a homologous dsDNA template, being the sister chromatid or the homologous chromosome, forming a joined DNA molecule, the D-loop structure. As meiosis progresses from leptotene to zygotene to pachytene homologs become synapsed with the ZYP1 protein installed as a transverse filament of the synaptonemal complex. DNA synthesis on the homologous matrix primed by the invasive 3’ strand creates extended recombination intermediates that can then be captured from the second end of the DSB to form the double Holliday junction (dHJ). Stabilization of these recombination structures involves proteins of the ZMM (ZIP1-4, MSH4/5, and MER3) pathway. The dHJ can be resolved as a class I CO or dissolved by the topoisomerase complex to form non-COs. An alternative pathway, which accounts for only 10% of the overall COs in Arabidopsis (called class II COs), relies on the resolution of the D-loop intermediates by structure-specific endonucleases involving MUS81. The recombination intermediates can also be resolved as non-COs upon unwinding of the extended invading DNA strand and reannealing to the complementary strand on the second end of the DSB following a repair mechanism called SDSA (Strand Displacement Synthesis Annealing). The dissociation of earlier invasion intermediates is performed by partially redundant anti-COs pathways involving notably the FANCM and RECQ4 DNA helicases [10,11]. If the local transfer of genetic information from the repair template to the broken molecule occurs, this may lead to gene conversion. In Arabidopsis, only 7–12 DSBs amongst the 100–200 formed at the leptotene stage of the prophase of meiosis are repaired as COs, the others being resolved as non-COs or by repair on the sister chromatid [8,12]. The mechanisms preserving CO number when the DSB number is modulated on one hand, and inhibiting new COs adjacent to existing CO, on the other hand, are called homeostasis and interference, respectively ([13,14] for recent reviews in plants). In contrast to class I COs, class II COs are not prone to interference.

Recent fine scale data in Arabidopsis, resulting from the sequencing of SPO11 oligonucleotides prepared from floral buds, indicate that hot spots (HS) of DSB formation occur preferentially in permissive, nucleosome-depleted regions like gene promoters, terminators, and introns where chromatin is accessible [15]. DSB HS are found in AT-rich sequences frequent in transcriptional start (TTS) and termination (TTS) sites and introns of genes that may contain DNA transposons. In maize, sequencing of ssDNA attached to RAD51 prepared from male meiocytes, which constitutes a proxy for identifying DSB sites, has also shown that DSB forms peaks around the TSS and TTS but are not specifically directed to sites of highly transcriptionally active genes [3]. In maize, 72% of DSB HS sites contain a 20 bp-long GC-rich, degenerated DNA sequence, also found at CO sites, and DNA methylation might be a regulator of HS strength [3]. Most DSBs occur in repetitive DNA and are associated with nucleosome-free chromatin. While in both species, a strong inhibition of DSB by DNA methylation is observed, the correlation of DSB HS and occurrence of the H3K4me3 active histone mark remains poor [3,15]. 

Coalescent population analysis of Arabidopsis historical COs in genetic resources has shown that COs are associated with active chromatin modifications including greater deposition of the histone variant H2-AZ and enrichment of the H3K4me3 mark, in regions of low nucleosome occupancy and low DNA methylation [4,16]. As mentioned above for DSBs, CO frequency is very high in proximity of TSS and TTS which are regions of low nucleosome occupancy facilitating Pol II transcription initiation and termination. Several DNA motifs (A–rich, CCN, and CTT repeats found upstream and downstream TSS) are enriched in CO regions [15,17]. While there is a genome-wide global positive correlation between SPO11 oligos and COs positions in Arabidopsis, the fine scale correlation becomes weaker, indicating the influence of other factors such as interhomolog polymorphism downstream of DSB formation for obtaining successful repair in CO [15]. In maize, while DSBs appear to occur genome wide, repair as CO is restricted to DSBs occurring in genes [3].

The presence of specific DNA shape structure and low CA dinucleotide frequency was found a predictor of CO occurrence specific to rice [18,19]. Other features favoring COs and shared with other plant species (maize, tomato, and Arabidopsis) encompass DNA helix twist, and AT, TA, AA, and TT dinucleotide frequencies. Rice genome topologically associated domains (TADs), defined as regions of high chromatin inter-connectivity have significantly higher SNP density and recombination rate compared to inter-TAD regions, but also surprisingly exhibit significantly higher CG and CHG DNA methylation and H3K9me2 levels, which are generally negatively correlated with COs [20]. Still, in rice, two historical fine scale recombination maps have been established in the japonica and indica groups using population SNP datasets and simulating coalescent events between variants derived from 150 re-sequenced genomes [21]. Similar to Arabidopsis, rice COs are preferentially found enriched in promoter regions, upstream TSS, downstream TTS while depleted in gene bodies, and associated with permissive histone modifications such as H3K4me3, but also here H3K9ac, H4K12ac, and H3K27me3. COs were also found associated with methylated CHG and CHH sites [21]. This coincidence of DNA methylation enrichment and high CO frequency in rice is contradictory to observations of hypomethylation at CO sites in Arabidopsis [15,17]. This discrepancy has been attributed to the use of epigenomic chromatin datasets from non-meiotic, somatic tissues which may not necessarily reflect the chromatin state during meiosis. The observation that a large set of transposable elements are expressed preferentially or specifically in meiocytes versus anthers [22] has indeed suggested that hypomethylation may occur at meiosis onset. However, in Arabidopsis, male meiocytes are found hypermethylated compared to leaves in the CG and CHG context but hypomethylated in the CHH context [23]. 

Taken together, these results suggest that the recovery of recombinants is influenced, first by the chromatin accessibility to the DSB formation and processing machinery, second, by the sequence divergence (polymorphism, structural variation) existing between the two parental DNA homolog molecules in the region where the DSB has occurred, that may lead or not to an abortion of the invasion of the homologous template, and third, by the DSB repair mechanisms that mainly restore the parental genotype [24,25]. Post-meiotic events can also limit recombinant recovery, for instance through the lethal action of gametophyte- or sporophyte- development genes conducting to the decay of some allelic combinations in gametes and zygotes, respectively, as illustrated in rice [26]. Hereafter, we will focus on the recent advances accomplished in the modulation of meiotic recombination (global stimulation and/or redistribution) and avenues open for the targeting of meiotic recombination at specific genome sites.

## 3. Stimulation and Redistribution of Meiotic Recombination

### 3.1. Modulation by Sex

Differences in the number and distribution of COs are observed in male and female meiosis, a phenomenon called heterochiasmy. Contrasted chromosome 4 genetic map lengths have been observed in Arabidopsis male and female meiosis, with lengths of 88 cM and 52 cM, respectively. Furthermore, an up to 4-fold CO increase over the average chromosome value is observed in the distal regions in male meiosis but not in female meiosis [27]. Analysis of the whole genome confirmed these dramatic differences in map length (575 cM versus 332 cM in male and female meiosis, respectively) and CO distribution patterns [28]. Increased CO occurrence in male meiosis has been reported in barley [29] and maize [30]. However, another study in maize contrastingly reported a similar number of COs in male and female meiosis [31] but converged to report a parallel distal increase in both the male and female chromosomal CO landscapes [30,31]. This apparent discrepancy between Arabidopsis and maize might be ascribed to the very contrasted CO distribution landscapes in the two plants.

### 3.2. Modulation by Environmental Conditions

Environmental factors such as temperature and nutrients have been reported to influence meiotic recombination, with maximum enhancements ranging from 20 to 30% ([32], for a review). The effect of temperature on CO frequency is complex, both moderately low and high temperatures had a promoting effect while lower and higher temperatures have a deleterious effect in impairing synapsis, conducting a meiosis failure and loss of fertility. Beyond influencing the overall number of CO, higher temperatures may alter their distribution, as exemplified in barley: a male meiosis-specific distal decrease and a shift for interstitial and proximal regions is observed on meiotic chromosomes of plants flowering at 25 to 30 °C compared to 15 °C [29,33]. In Arabidopsis, both high and low temperature increase meiotic COs through additional class I COs, reflecting again a non-linear U-shaped dynamics across a moderate T° (12–29 °C) range [34,35]. An increase in length of the synaptonemal complex, a component of which (ZYP1) is known to be involved in the control of CO interference [36], is associated with this increase of CO frequency observed at higher temperatures in barley [33] but not in Arabidopsis [34]. Temperature is also known to alter the chromatin state since a concurrently increased deposition of the permissive H2-AZ variant deposition and CO frequency is observed in plants flowering at low (12 °C vs. 21 °C) temperature, confirmed by the absence of such an effect in an H2-AZ deposition mutant [4]. Taken together, these results demonstrate that modulation of temperature could be used as a factor to—albeit moderately—stimulate recombination. 

### 3.3. Modulation by Novel Genomic Situation

Specific genomic situations, such as allo-triploidy in Brassicaceae, have been found to stimulate meiotic recombination [37] and redistribute it notably in heterochromatic regions [38], likely through an epigenetic mechanism. AAC allotriploid plants, resulting from a cross between *Brassica napus* AACC, 2n = 4x = 38 and *Brassica rapa* CC, 2n = 2x = 18, exhibit up to a 3.6-fold higher number of COs along the entire A chromosomes compared to diploid AA or allotetraploid AACC hybrids. This stimulation is associated with a dramatic change in the recombination distribution that occurs in the vicinity of centromeres, normally deprived of COs in diploids, and is mostly due to an increase in class I CO and consequently to a reduction of interference [38]. While interesting in the Brassicaceae case study, implementation of the triploidy strategy in other species appears more difficult.

### 3.4. Modulation by Epigenetic Factors

Experimental evidence that epigenetic factors influence recombination has been provided by analysis of DSBs HS and CO sites in mutants of genes involved in the maintenance of CG (*MET1*) and non-CG (*CMT3*) DNA methylations or imposition of repressive H3K9me2 marks (H3K9 methyltransferase genes *KYP/SUVH4 SUVH5 SUVH6*) [39,40,41,42]. Global loss of cytosine methylation in *Atmet1* increases CO formation in heterochromatic and centromeric regions and concurrently decreases them in pericentromeric regions without modifying the overall DSB number [39,40,43]. On the other hand, loss of methylation at non-CG, CHG sites through mutation in the DNA methyltransferase *AtCMT3*, increases DSBs and COs within pericentromeric regions [42]. Beyond DNA methylation, histone modifications and histone variant deposition, other layers of epigenetic control of DSB induction and repair may exist. Deep sequencing of total small RNAs from leaves and meiocytes has been recently conducted in an Arabidopsis *spo11-1* mutant [44]. AtSPO11-1–dependent meiocyte sRNAs enrichment at meiotic recombination associated DNA Motifs (CTT-repeat motif associated with genic regions and A-rich motif associated with gene promoters) suggest that AtSPO11-1–dependent sRNAs tend to be associated with the open chromatin structure, which might favor meiotic recombination HS, indicating a putative role of sRNAs in meiotic recombination. Taken together these results show that changes in meiotic recombination distribution, albeit moderate, can be expected from the inactivation of genes involved in the maintenance of CG and non-CG DNA methylation in plants. However, the essential feature of these mechanisms makes their inactivation accompanied with pleiotropic, detrimental effects on the plant phenotype that obviously limits its use in breeding. In that perspective, either down-regulating these genes specifically during meiosis or identifying genes specifically involved in DNA methylation maintenance during meiosis would be of great interest.

### 3.5. Modulation by Altering the Expression of Genes Involved in Homolog Synapsis and DNA Repair

Modulation of CO frequency and distribution in plants has proven possible through the inactivation or overexpression of genes involved in the regulation of meiotic recombination pathways, notably in the homologous chromosome synapsis, invasion, and resolution of recombination intermediates steps (Figure 2). According to the gene, species-specific effects and/or differential impacts on plant fertility have been observed. 

Partial inactivation of the transverse filament protein of the synaptonemal complex, *ZEP1*, which is the rice ortholog of Arabidopsis *ZYP1*, allowed a 1.8-fold increase in class I CO. However, the complete inactivation of *ZEP1* conducted to full male sterility, while maintaining female fertility [45,46]. In Arabidopsis, the absence of ZYP1 abolishes homologous chromosome synapsis and increases the number of COs by 54% by suppressing the class I CO interference phenomenon and heterochiasmy, but by contrast with rice, no consequence on fertility [36].

Inactivation of the AAA-ATPase *FIGL1*, which is an antagonist of the RAD51 and DMC1 recombinases and thereby limits the homologous molecule invasion step, has been shown to increase recombination only moderately [47]. However, *figl1* exhibits a strong synergistic action with the concurrent inactivation of the DNA Helicases *RECQ4a/b* in Arabidopsis, showing a highest 8–10-fold increase in COs [48]. The *figl1* mutant proved to be sterile in certain crops [49,50]. The respective inactivation of the *Fanconi anemia of complementation groupM* (*FANCM*) helicase and of the *slow growth suppressor 1 (SGS1)/Bloom syndrome protein (BLM)* homologs *RECQ4a/b* involved in the dissolution of recombination intermediates and their resolution as non-CO in the non-interfering class 2 CO formation pathway, increases recombination in Arabidopsis pure lines by a factor of 3 and 4 [48]. However, while the *recq4a/b*-mediated increase is maintained in polymorphic crosses, the *fancm*-mediated effect is no longer observed [48]. The ectopic expression of the pro-crossover E3 ligase protein of the ZMM pathway *HEI10* [51] has a CO-stimulating effect which was recently found to act additively with *recq4a/b* to further enhance the frequency of recombination in Arabidopsis from 7.5 to 31 COs per F2 individual [52]. In crops, while the *fancm* mutation produced the expected stimulatory effect in oilseed rape meiotic CO frequency [53], it led to altered fertility or sterility in pea, tomato, and lettuce [49,54]. In rice, the inactivation of *FANCM* in a lowly polymorphic intra-japonica hybrid (1SNP/10Kbp) resulted in a 2.3x stimulation of recombination, without significant reduction of fertility [49]. On the other hand, the inactivation of the *RECQ4* DNA helicase increased recombination by 3–4-fold with no concomitant fertility alteration in crops, including rice, pea, and tomato [49]. This positive result has been recently extended to a distant homeologous context in tomato [55] and to barley [56]. The inactivation of the DNA helicase RECQ4, therefore, appears to be a universal tool for stimulating meiotic recombination at the genome scale in crops. 

However, as the *recq4*-stimulating effect appears locally inhibited by the level of interhomolog sequence polymorphism [49], it may not result in a redistribution of CO events towards highly polymorphic regions, which are generally depleted in COs. Analysis in several Arabidopsis recombinant populations have indeed established an initial positive correlation of CO frequency with increasing SNP density, that relationship becoming negative when a certain divergence threshold has been reached [57]. As mentioned earlier, DSB repair occurs through the homology-driven invasion step of the ssDNA into a homologous molecule, sister chromatid, or homologous parental chromosome, then used as a template for DNA synthesis. When too divergent, the detection of base-pair mismatches is thought to dissociate the early recombination intermediates thus preventing the risk of non-allelic COs and genome rearrangements. In yeast, the MMR proteins play an important role in preventing recombination between such divergent sequences [58]. Notably, MSH2, contained in mutS related heterodimers, acts as an anti-recombinase upon the action of the SGS1 helicase, promoting disassembly of mismatched D-loop intermediates [59,60]. In Arabidopsis, *AtMSH2* has been found to affect homologous recombination as a function of sequence divergence and notably displays an anti-recombination meiotic effect [61]. Unexpectedly, mutation of *AtMSH2* in Arabidopsis did not increase COs in polymorphic regions such as peri-centromeres but enhanced COs in less polymorphic, sub-telomeric regions pointing for a pro-crossover role of AtMSH2 towards more polymorphic regions [57]. Another protein essential in the regulation of meiotic recombination, recently discovered in rice, MEICA1 (meiotic chromosome association 1), has been suggested to be another anti-CO factor preventing non-allelic CO, through its interaction with the topoisomerase TOP3α and the plant-specific MutS protein MSH7, the Arabidopsis MSH2-MSH7 complex having the capacity to recognize specific mismatches [62]. Along the same line, the *TaMSH7* copy located on chromosome 3D of bread wheat was recently identified as a key regulator of homeologous recombination, providing novel opportunities to enhance alien gene introgression in this crop [63]. 

Finally, a member of the RPA heterotrimeric (RPA1, RPA2, and RPA3) protein complex is well known for its role in protecting the ssDNA ends notably those exposed following the endo-nucleolytic cleavage of SPO11 oligonucleotides, RPA1a, has been found essential for limiting chiasma formation. It has been suggested that RPA1, acts on Class II COs through an interaction with the FANCM-BTR complex in the processing of recombination intermediates, indicative that RPA1 might be a putative new target for unleashing COs in crops [64].

### 3.6. Interest in Stimulating and Redistributing Recombination in Breeding

The consequences of achieving increased recombination mediated by inactivating the anti-CO gene *RECQ4* or implementing the triploidy-based strategy over successive cycles of recurrent selection have been theoretically simulated in both rice and *B. rapa* [65]. It was assumed that *recq4* stimulates COs while maintaining their overall distribution shape whereas triploidy both stimulates COs and redistributes them in pericentromeric regions in both species. Increased recombination was found to improve response to selection and to enhance the genetic gain by up to 30% after 20 generations, with the visible effect observed after 4–5 generations. The genetic gain was larger with the second strategy that includes CO redistribution, in both *B. rapa* and rice, though, as mentioned previously, triploidy cannot be implemented in rice. 

Enhanced recombination can also narrow down the size of chromosomal fragments of the donor parent in the recurrent parent, enhancing QTL mapping precision and facilitating trait introgression without linkage drag. Smaller (7.6 Mb vs. 16.9 Mb) and more numerous (21 vs. 9) introgressions of *B. rapa* occurred in AAC hybrids compared to AACC allotetraploid hybrids, indicating that the stimulation of recombination is also efficient to precisely map QTL carried in cold regions of the oilseed rape genome. Allotriploid AAC hybrids are therefore highly efficient to introduce novel variations within oilseed rape varieties [66]. 

While an overall stimulation of recombination in elite materials might not be of interest for breeders since it may also disrupt beneficial linkages, redistributing COs in the large regions of crop genomes deprived of recombination would be of great value. Integrating these tools in recurrent populations and pre-breeding generations for diversity shuffling appears the most appropriate and promising.

## 4. Targeting of Meiotic Recombination

### 4.1. Targeting Somatic COs with CRISPR/Cas9

In addition to targeted site mutagenesis, it has been demonstrated that the induction of DSBs at a determined site of the genome by CRISPR/Cas9 can result in COs in somatic cells. This method has been used to refine genetic maps in yeast [67]. In tomato, the induction of double-strand breaks by CRISPR/Cas9 at a fruit staining locus to identify recombinants led to a high frequency of somatic COs, a small fraction of which have been integrated into the germline and transmitted to the progeny [68]. CRISPR/Cas9 induced inter-homolog recombination events in both euchromatic and heterochromatic regions of Arabidopsis chromosomes mainly produce few bp to few kbp gene conversions that are transmitted to progeny but rarely COs [69]. More recently, a constitutively expressed and guided Cas12a (Cpf1) was also found able to drive targeted CO at two different loci in somatic maize cells that were inherited in the next generation [70]. According to the transformation event, the targeted CO was detected in only a subset of seeds or in 100% of progeny seeds indicative of different time courses of induction of somatic CO during primary transformant development.

Though systems for germline- or meiosis-specific expression of CRISPR/Cas9 exist [71,72], targeting CO by CRISPR/Cas9 specifically during meiosis has not yet been reported. This is presumably due to the need for the presence of SPO11 and other associated proteins of the catalytic complex at the DSB site for proper endo-nucleolytic cleavage, resection, and processing, that eventually lead to a repair as meiotic CO. An alternative is, therefore, to use partners of the natural catalytic complex itself to promote the induction of a DSB at a specific site in the genome. 

### 4.2. Targeting Meiotic COs with Partners of the DSB Catalytic Complex

The evolutionarily conserved SPO11 proteins share sequence similarities with the A subunit of the archaeal Type II DNA topoisomerase VI (TOPO VI A) [73]. In archaea, TOP VI acts as a heterotetramer comprising two A subunits that cleave DNA and two B subunits that use ATP binding and hydrolysis to coordinate the passage of the DNA duplex through the DSB (recent update in [9]). Spo11, which derives from the DNA-cleaving Topo VI A subunit, forms a dimer to induce DSBs in yeast [74]. In yeast, the formation of the Spo11 core complex, also comprising Rec102, Rec104 and Ski8, has been structurally and functionally characterized recently [75]. In contrast to yeasts and mammals, two distinct Spo11 paralog proteins, AtSPO11-1 and AtSPO11-2, exist in Arabidopsis and most plants [76], and likely associate to induce meiotic DSBs [77,78,79]. AtSPO11-1 and AtSPO11-2 interact with a meiotic TOP VI B (M-TOPVIB), which has structural similarities with the ancestral TOPO VI B subunit, to assemble into a hetero-tetrameric complex [80].

In yeast, the fusion of the Gal4 DNA binding domain (GAL4BD) with SPO11 was sufficient to locally increase meiotic DSB formation and recombination (COs and gene conversion) at GAL4 binding sites located in naturally cold regions such as in the GAL2 promoter [81,82]. High resolution mapping of the induced DSBs showed that they occur approximately 20 nt from the Gal4 binding sites with a slight sequence preference [83]. Genome-wide analysis of the GAL4BD-SPO11 chromatin-associated sites and DSB formation in yeast have shown that the binding of the SPO11 fusion to target DNA sequences in cold regions is not always sufficient for triggering DSB formation [81]. More recently, the GAL4BD:SPO11 fusion strategy has been expanded to include other DNA binding modules such as full-length transcription factors, synthetic zinc fingers, and transcription activator-like effector (TALE) in yeast [84]. Similarly, the potential of expression of a nucleolytically inactive Cas9 (dCas9) SPO11 fusion (dCas9::SPO11) guided to specific genomic sites by one or several guide RNAs (sgRNA) has been explored. Targeted DSB formation and a 2- to 6-fold stimulation of meiotic crossover recombination have been observed over a panel of target regions in a spo11∆ background [84]. 

In plants, a dCas9::mTOPVIB fusion protein has been accumulated under the control of the *mTopVIB* promoter in a null *mtopVIB* mutant of Arabidopsis for targeting DSB at a CO HS locus via the simultaneous expression of 6 sgRNAs. However, this did not significantly enhance the local meiotic CO frequency, possibly because the frequency was naturally high at this locus [85]. The interactions existing between the members of the catalytic subcomplex (AtSPO11-1, AtSPO11-2, mTOPVIB) with the other DSB proteins, belonging either to the RMM-like complex (namely DFO, PRD2/AtMEI4, pHS1/Rec114), ensuring the anchoring of the catalytic complex to the axis, or to the DSB resection complex (COM1, RAD50, MRE11, and NBS1) through an interaction with PRD1 itself interacting with PRD3/MER2, have been recently clarified in Arabidopsis [86]. Further studies may provide alternative strategies for targeting recombination with dCas9. Whether the dCas9-mediated cargo of several key proteins of these complexes to the desired cleavage site appears necessary, the recently established Suntag or aptamer technologies could be implemented (recently reviewed in [87]). 

### 4.3. Interest in Targeting Recombination in Breeding

The main interests in directing COs to specific sites are to break unfavorable linkage and to increase genetic gain. When neighbor genes residing in regions with low recombination contain, respectively, favorable and favorable alleles, introgression of a favorable allele will most of the time be accompanied also by an unfavorable allele at the other genetically linked locus This phenomenon, known as linkage drag in breeding, conducted for instance to the introduction of the unfavorable root system and drought susceptibility in modern cultivars of bread wheat and rice, respectively [88,89]. Other examples of tight detrimental linkages in rice include QTLs for grain weight and grain number [90] and a blast resistance gene and a QTL conditioning spikelet fertility [91].

The potential for increasing the genetic gain in targeting recombination has been simulated: when targeting one CO per chromosome in maize the expected predicted gain for yield and agronomic traits could be on average doubled, this advantage ranging from 105 to 400% according to the considered population and trait [92,93]. For barley, a self-pollinated crop, prediction models have shown that targeting recombination at a single site on either chromosome 2, 4, or 7 could increase the relative genetic gain by 118% while a simultaneous targeting of chromosomes 2 and 4 may increase it up to 128% [94]. 

Targeting meiotic recombination is, therefore, a highly desirable goal in breeding, as it would allow the breeder to disrupt undesirable associations, capture genetic diversity in “cold” genomic regions, and accelerate the introgression of new allelic combinations without compromising patiently established beneficial allelic associations at other essential loci.

## 5. Conclusions and Prospects

The manipulation of meiotic recombination has attracted a lot of academic and industry interest in the last few years [95]. The past and ongoing efforts to decipher the function of the genes that regulate meiotic recombination between homologs and homeologs have opened avenues to enhance and redistribute meiotic recombination in plants. Progress in the understanding and manipulation of CO interference could lead to imposing the strict occurrence of a single CO per homolog pair, which would allow the creation of fertile tetraploids in diploid crops [96]. Chromosome engineering by CRISPR/Cas is also a promising technology for re-establishing COs in naturally depleted chromosomal regions, especially where large non-recombining inversions exist, as recently exemplified in Arabidopsis [97].

Shuffling diversity through stimulation of recombination in pre-breeding generations, for instance in recurrent breeding populations, via the inactivation and ectopic expression of anti- and pro-cross-overs genes, respectively, and without compromising plant fertility, should prove feasible. Targeting meiotic recombination would certainly be the ultimate tool to assist the breeding process with speed and accuracy, accelerating the genetic gain. Again, a better understanding of the mechanism of meiotic DSB induction, as well as the processing machineries and factors modulating their efficiency in plants, will be necessary to cargo a suitable protein complex to the desired site(s) and thus to achieve programmed homologous recombination in crops.

## 6. Patents

The Meiogenix company holds the patent on targeted meiotic recombination.

## Figures and Tables

**Figure 1 biology-11-00369-f001:**
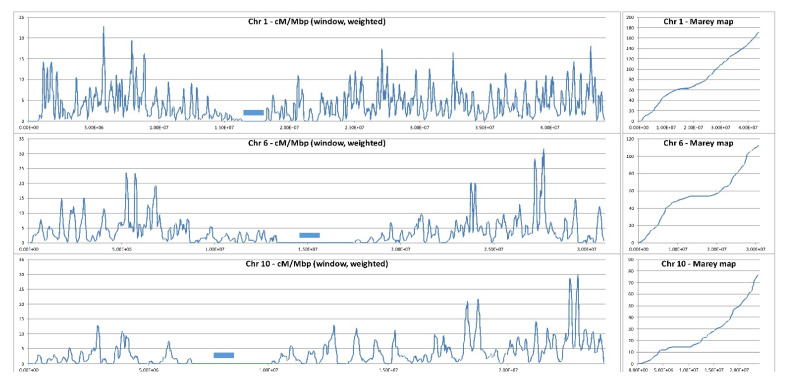
Local recombination rate along three representative chromosomes of rice. Cross-over positions have been deduced from low coverage (4x) Illumina reads of 380 F2 progeny plants derived from selfing of an indica/japonica hybrid. X axis: chromosome coordinates (Kitaake genome) in base pairs. Y axis: local recombination fraction in centiMorgans per megabase (cM/Mb). Blue bars mark centromere positions (Courtesy of Mathias Lorieux, IRD, France).

**Figure 2 biology-11-00369-f002:**
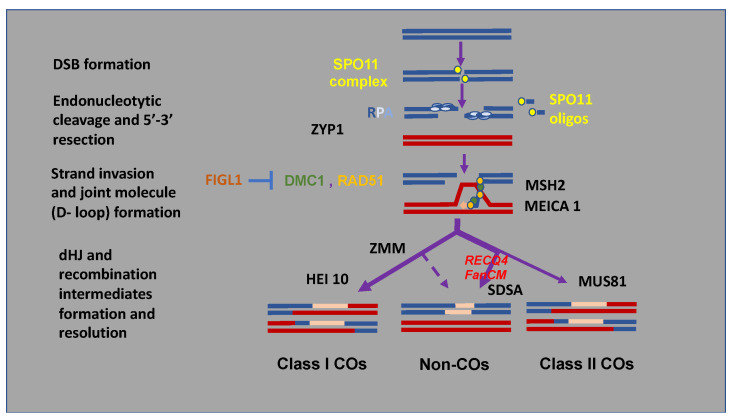
A simplified view of pathways for repairing Meiotic Chromosomal Double-Strand Breaks (DSB) and their modulation by altering key regulatory genes. Meiotic recombination is initiated within the mother cells of microspores and megaspores by the self-induction during the prophase of the first division of meiosis of several hundreds of DNA DSBs by the SPO11 complex. Occurrence DSBs trigger the pairing of homologous chromosomes (blue and red bars). In the current model of DSB-associated DNA synthesis, the CO formation pathway proceeds mainly in repairing the DSB by homologous recombination using an intact chromatid of the paired homologous chromosome as a template. This pathway is initiated by the invasion of the 3′ end of a single end of the DSB which initiates DNA synthesis on the homologous DNA molecule. The capture of the second end of the double-stranded break facilitates further synthesis using the 3′ end on the other side of the break. A specific joint molecule structure linking the 4 strands of the 2 homologous chromosomes, the double Holliday junction (dHJ) is formed. The resolution of this structure usually leads to a reciprocal exchange of large DNA segments between two chromatids of the homologous chromosomes, the cross-over. This major pathway (class I) accounting for 90% of CO formation is called the ZMM (*ZIP1-4*, *MSH4/5,* and *MER3*) pathway. Other repair pathways pass through a variety of recombination intermediates that may mature in a minority way (10% in Arabidopsis) in so-called class II COs through MUS81 but mainly form non-COs, which can lead to gene conversions. Neosynthesized DNA appears in pink.

## Data Availability

Not applicable.

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
