# Peer review of "Manipulation of Meiotic Recombination to Hasten Crop Improvement"

_biology, 2022, doi:10.3390/biology11030369_

Round 1

Reviewer 1 Report

The review describes the state of the art regarding the various ways of manipulating the meiotic recombination pathways to enhance or redistribute crossovers in plants. The review is well organized and gives a good overview of the strategies that could be envisioned to achieve the goal. The references are appropriate.

However, the figure 1 and its legend need to be revised before publication:

. there is no reason to draw ZYP1 at the DSB steps. This figure that does not take into account the various elements of the axes does not need ZYP1 drawing as the appropriate stage to draw it does not exist

. in the legend, synapsis should not appear between brackets after the word pairing as it suggests that the two are synonymous. But they are not! Synapsis implies the formation of the synaptonemal complex whereas pairing is the alignment of the two homologs. One precedes the other. And pairing can occur without synapsis: for example, in zyp1s mutant in Arabidopsis, DSBs are formed as in wild type, pairing occurs as in wild type, but there is no synapsis and there are more COs….

This review will constitute a good paper to be given to people who starts in the field

Other minor points:

  • Line 76-77: The fact that in Arabidopsis CO frequencies are higher in the pericentromeric regions is only true for female meiosis. In male meiosis as mentioned lane 207, the CO pattern follows a U shape with high CO rates in distal and in pericentromeric regions.
  • Line 127: it is the dissolution of the dHJ by the topoisomerase complex that produces the NCOS. The sentence should be modified accordingly
  • Line 137: the range of COs in Arabidopsis is more 7 to 12 than 5 to 10
  • Line 180: To avoid confusion, the two words “Hot Spot” should be used only for DSB and not for COs. It is well described in this review that the two do not coincide at 100% and hot spot has been first used in S. cerevisiae to describe the  regions that initiate meiotic recombination
  • Line 188: In Arabidopsis, Walker et al. have shown that DNA was hypermethylated in meiocytes compared to leaves in the CG and CHG context but hypomethylated in the CHH context. This work should be added.
  • Line 264: The word dramatic is not appropriate. Only small effects have been detected which are not found in all chromosomes. Please modify
  • Line 283: why is it surprising that zyp1s mutant are fertile in Arabidopsis? It is known that high CO rates do not affect fertility. Isn’t it the fact that zep1 mutant  is sterile in ricewhich is surprising ?
  • The authors should go carefully through the manuscript to put in italics the mutants, upper case the proteins and genes as sometimes it is difficult to know what they refer to.

Reviewer 2 Report

This review not only updates recent spectacular progresses in the mechanisms underlying meiotic recombination in both models and crops, but also surveys the new achievements in the manipulation of meiotic recombination in plants. Especially, authors point out potential applications of meiotic recombination mechanisms in fastening the crop improvement, which will meet the challenges of deploying crops with climate instability, resistant to pathogens and pests, and sparing in input requirements. This is a very meaningful review. It provides us with a novel view about how to improve crop breeds with combining of gene editing techniques and manipulation of meiotic recombination. There are only some minor format concerns with me  in some paragraphs of this manuscript, line 344, 345, 369, 370, 383, 384, 424, 425, 447, 456, and 466.   
